# Peer Status Influences In-Group Favoritism in Pain Empathy During Middle Childhood: Evidence from Behavioral and Event-Related Potentials Studies

**DOI:** 10.3390/brainsci14121262

**Published:** 2024-12-16

**Authors:** Yiyue Chen, Jingyuan Liang, Gaoxin Han, Xue Yang, Juan Song

**Affiliations:** Faculty of Psychology, Tianjin Normal University, Tianjin 300387, China; 2211340143@stu.tjnu.edu.cn (Y.C.); 2230350277@stu.tjnu.edu.cn (J.L.); 2310340018@stu.tjnu.edu.cn (G.H.); 2210340004@stu.tjnu.edu.cn (X.Y.)

**Keywords:** children, empathy for pain, peer status, in-group favoritism, ERP

## Abstract

Background/Objectives: Empathy for pain enhances our ability to perceive pain and recognize potential dangers. Empathic bias occurs when members of the in-group evoke more intense empathic responses compared to out-group members. In the process of interacting with peers, children develop peer status and spontaneously form peer groups. The present study examined how peer status affects pain empathy in mid-childhood individuals. Methods: A behavior and an event-related potential (ERP) study were conducted. Participants were exposed to pictures of different peers in painful or non-painful situations and completed the pain and unpleasantness rating tasks. Four types of peers were included: popular, rejected, neglected and unfamiliar peers. Results: The behavioral results suggested that the influence of peer status on cognitive empathy is more salient, and the empathic response to unfamiliar peers is higher than neglecting and rejecting peers. The ERP results indicated that larger P3 and LPP amplitude were observed in the painful stimulus condition than in the non-painful stimulus condition. The findings also showed that the popular peers elicited larger LPP amplitude than other peers. The LPP response to unfamiliar peers was larger than to neglected peers. Conclusions: All these results demonstrated that mid-childhood individuals showed empathic bias to in-group members, but it was influenced by peer status in the cognitive processes of pain empathy.

## 1. Introduction

Empathy for pain encompasses the perception, understanding and response to the pain of others, involving emotional, cognitive and motivational dimensions [1,2]. It is a vital human attribute for social interaction, facilitating the adoption of others’ emotional states while maintaining a distinction between the self and others [3]. Researchers widely acknowledge that pain empathy generated by individuals encompasses both bottom-up affective empathy, which involves spontaneous or vicarious emotional experiences in response to others’ feelings, and top-down cognitive empathy, which enables the recognition of others’ emotions and comprehending their perspectives [4]. However, the relative weight of these two components may vary across different contexts, and they complement each other to maximize an individual’s social adaptation [5].

During interpersonal interactions, individuals often categorize people into in-groups and out-groups [6]. The sense of group identity leads to more positive evaluations of in-group members, resulting in in-group favoritism [7]. Social group information is an essential cue and indicator in social interactions, and empathy is often influenced by the identity of group members [8]. Specifically, witnessing in-group members experience pain tends to elicit more intense empathic responses [9,10], a phenomenon researchers refer to as empathic bias [11].

Previous studies have used functional magnetic resonance imaging (fMRI) to explore the impact of social group membership differences on pain empathy. Researchers have found that observing pain in individuals of the same race leads to greater activation of the anterior cingulate cortex (ACC) [12]. The study by Cao et al. [13] further found that observing pain experiences in individuals of other races resulted in a significant increase in activation of the ACC as exposure to other races increased. Studies of strangers and lovers have found similar results. The pain experienced by the lovers activated the anterior cingulate gyrus and insula more significantly [14]. Hein et al. [15] conducted a study of fans of different teams and showed that seeing fans of the same team experience pain resulted in greater activation of the left anterior insula. The results of a meta-analysis showed that the left cingulate gyrus and left insular lobe responded to the in-group more than the out-group response, which may reflect the individual’s greater empathic attention to the pain of the in-group members and the painful facial expressions of the in-group members may be more easily encoded during empathic processing [16].

Event-related potential (ERP) technology overcomes the low temporal resolution of fMRI and can reflect the temporal differences between cognitive and affective empathy. Previous ERP studies on pain empathy have primarily focused on components such as the N100 (N1), N200 (N2), P300 (P3) and the late positive potential (LPP). It is widely posited that ERP components within the 100–300 ms time windows, such as N1 and N2, are associated with affective sharing and perception processes [17] and are not influenced by cognitive evaluative processes [18], reflecting affective empathy. Components that occur after 300 ms, such as P3 and LPP, indicate how individuals evaluate painful stimuli—a process influenced by attention that reflects cognitive empathy [19]. Research on empathy for pain has examined how social group membership affects both the early automatic processes and the later controlled processes, though findings have been inconsistent. Some research suggests that social group only influences the early processes of pain empathy. Sheng and Han [20] found that the pain faces of in-group members elicited larger N2 amplitudes, with no similar results observed for the late component P3. Additionally, participants did not show racial preferences in implicit and explicit attitude tests. Researchers suggested that such racial preferences might be generated unconsciously. Another research found that black faces elicited larger N1 amplitudes in white participants, while white faces elicited larger N2 amplitudes [21]. However, some studies showed that social group membership influences both the early affective perception and later cognitive evaluation processes of pain empathy [22]. Song et al. [23] collected and analyzed ERP components in friend and stranger priming conditions. During the pain judgment tasks, the N1 over the frontal-central region was larger under the stranger priming conditions, but the P3 over the central-parietal region was smaller under stranger priming compared to friend priming. In another study, participants exhibited larger N1 and N2 amplitudes in response to friends’ pain, while strangers’ pain resulted in greater LPP amplitudes [24]. Therefore, the impact of social group membership on different processes of pain empathy needs further investigation.

Although in-group favoritism can occur in any type of social group [16], most studies have focused on groups with clear boundaries, such as race, gender and age, including some artificially created and differentiated small groups. For example, groups are formed based on clothing color, preferences for a particular painter or estimates of the outcomes of coin flip [25]. There are distinct rules for demarcating social groups, whether naturally established or artificially prescribed. However, in daily life, many informal groups form spontaneously and lack clear boundaries. People might know members of these groups or even have close relationships with them. Whether the pain experienced by members of these groups also triggers in-group favoritism remains to be discussed.

Previous research on children’s social group and empathy for pain is limited, but existing studies suggest that children’s preference for their social group develops with age and appears early in life. Infants prefer to observe and interact with people who are of the same race, language and preferences [26,27,28]. Children of preschool age also demonstrate in-group favoritism. Five-year-old children exhibit more positive evaluations of in-group members [29] and tend to select in-group members as friends or playmates [30]. In-group favoritism is evident in the preschool age but diminishes in middle childhood. Aboud [31] suggested that children’s limited cognitive abilities hinder their capacity to consider multiple classifications simultaneously before middle childhood based on Piaget’s theory. Initially, children focus on themselves, then on groups and later on individuals in the groups, which shapes their attitudes toward different groups. After the age of eight, children are more likely to differentiate individuals within groups. This increased attention to individual attributes within a group suggests a decrease in-group-based prejudice, meaning that in-group favoritism typically emerges in preschool and gradually declines after middle childhood. The study by Jordan et al. [32] supports this perspective. They found that 6-year-old children exhibit in-group favoritism, whereas eight-year-old children do not display an obvious preference for the in-group or out-group. Research on Chinese children indicated that middle-grade students (with an average age of 9.122 and 9.950) possessed a more pronounced sense of group boundaries and manifested marked in-group favoritism, whereas high-grade students (with an average age of 10.707 and 12.000) did not exhibit in-group favoritism [33]. However, some research suggests that group preference persists throughout childhood. For example, Susskind and Hodges [34] recruited 67 children 9 years and 10 months and found that these children showed in-group favoritism and out-group derogation. In the study conducted by Gonzalez-Gadea et al. [35], the 4~6 years age group, 7~8 years age group and 9~11 years age group all showed significant in-group favoritism. Besides, in Buttelmann and Böhm’s [28] study on 6- and 8-year-old children, the 8-year-olds not only exhibited significant in-group love but also demonstrated out-group hate. Therefore, this study aims to explore whether children in middle childhood show a preference for in-group members.

Schools are the primary places for children’s learning and socialization, where they spend a considerable amount of time interacting with peers. The collective activities of children of similar ages constitute peer relationships. Interactions and cooperation with peers affect children’s physical, mental and social development [36]. The importance of peer relationships to children is changing with age. Therefore, it is crucial to investigate how children’s peer relationships impact their ability to empathize with others’ pain in a school setting. Peer status greatly affects children’s mental health. Peer status refers to an individual’s position within a group of children who interact with each other, are of similar age and have similar cognitive abilities. The peer nomination, widely utilized in sociometric methods, assesses peer status by asking children to nominate the three classmates they like the most and the three they like the least to play with. Social preference and social impact scores are calculated, and children are categorized into popular, rejected, neglected, controversial and average children [37]. Popular and rejected children refer to those who are liked or disliked by most peers, respectively; neglected children are those who receive few positive and negative nominations; controversial children are liked by some children but considered disruptive and disliked by others; average children do not fall into any of the above categories. Different types of peers often form naturally in children’s daily lives and are widely accepted and recognized by children, without obvious cues of group relationships. The peer nomination method allows people to assess the degree of popularity or unpopularity of children within a group, as well as their levels of peer acceptance or rejection. This method can also be utilized to evaluate the effectiveness of behavioral interventions [38].

Although peers belong to the in-group members of children and are more likely to elicit children’s in-group favoritism compared to unfamiliar peers, children have different attitudes and reactions to peers of different status. Whether and how different peer statuses affect in-group favoritism in empathy for pain remains uncertain. Rejected and neglected peers are not welcomed in the peer group, but they are still quite different from unfamiliar out-group members. Firstly, children spend more time with these two types of peers than with unfamiliar peers. They are more familiar with rejected and neglected peers, so the empathy they generate should also be higher [39,40]. Secondly, rejected and neglected peers may carry some unpleasant memories or emotions for children [41]. Children who have a good peer status typically maintain positive and close interactions with their peers. However, rejected peers often display aggressive behavior, while neglected peers usually play alone and participate less in social interactions [42]. Thus, they are more likely to become targets of bullying and exclusion [43]. Focusing on the potential impact of peer status on empathy for pain can help reduce school bullying, promote children’s mental health and create a positive and healthy campus environment. Therefore, we aimed to investigate the influence of peer status on empathic processes.

In summary, existing research has focused on adults and found in-group favoritism in empathic processes. However, the influence of social group membership on children’s pain empathy requires further investigation. Furthermore, there is an ongoing debate about children’s in-group favoritism, specifically focusing on whether children in middle childhood exhibit in-group favoritism. Factors that may influence children’s in-group favoritism on the empathic processes also need further exploration. Previous research examining in-group favoritism in pain empathy often categorized groups based on race, age and gender or artificially distinguished groups according to specific preferences or characteristics [25]. Children interact more frequently with their peers in their everyday lives. Peer status is a common and recognized way of classifying groups among children during their interactions with peers. This study focuses on whether peer status influences children’s in-group favoritism in empathy for pain. This study focuses on children in middle childhood and conducted two experiments. Experiment 1 investigated how children rate popular, rejected, neglected and unfamiliar peers in pain and unpleasantness intensity rating tasks to explore the influence of peer status on cognitive and affective empathy from a behavioral perspective. Experiment 2 used the event-related potential (ERP) technique to examine children’s electrophysiological responses to pain empathy for different peers, exploring whether peer status influences neural activity during both early automatic processing and later controlled processing of pain empathy. We hypothesized that in the painful condition of experiment 1, children would give the highest ratings to popular peers in the pain and unpleasantness intensity rating task. The ratings for unfamiliar peers would be higher than for neglected peers but higher ratings than for rejected peers. Besides, in the painful condition of experiment 2, the early components N1 and N2, as well as the later components P3 and LPP, would show the greatest amplitudes for popular peers, while the amplitudes for unfamiliar peers would be smaller than those for neglected peers but larger than those for rejected peers.

## 2. Experiment 1

### 2.1. Methods

#### 2.1.1. Participants

G*Power 3.1 [44] was used to estimate the required sample size. It was calculated that 23 participants were needed for the experiment (effect size f = 0.25, α = 0.05, 1 − β = 0.95). Forty fourth-grade elementary school students (20 boys, 20 girls) from four classes were enrolled for Experiment 1. Due to extreme values in the results, one girl’s data was excluded from subsequent analyses. Data from 39 participants were retained (20 boys, 19 girls, *M* = 9.55 years, *SD* = 0.64 years). All participants were native Chinese speakers with normal or corrected vision. They received a gift after the experiment. This study was approved by the Ethics Committee of the Academy of Psychology and Behavior, Tianjin Normal University.

#### 2.1.2. Design and Material

A 2 stimulus type (pain stimulus, non-painful stimulus) × 4 peer relationship type (popular peer, rejected peer, neglected peer, unfamiliar peers) within-subject design was employed.

Before the formal experiment, a limited nomination method was used in the four classes from which the subjects originated. Children were asked to write down the names of 3 classmates they would most like to invite to their birthday party and 3 classmates they would least like to be grouped with during class activities [45]. Positive and negative nomination counts were obtained and standardized within the class, and Z-scores were calculated. All children in the class got a positive nomination score (PN) and a standardized negative nomination score (NN). Besides, social preference (SP, positive nominate scores minus negative nominate scores) and social impact scores (SI, positive nominate scores plus negative nominate scores) were also calculated. The previous study [37] has developed a formula to classify children into sociometric groups: popular (SP > 1, PN > 0, NN < 0), rejected (SP < −1, PN < 0, NN > 0), neglected (SI < −1, PN < 0, NN < 0), controversial (SI > 1, PN > 0, NN > 0) and average (Not in previous categories). Controversial children and average children were excluded from this study. The names of unfamiliar peers were selected based on previous research [46], and participants were asked to assess their familiarity with the selected names using a 5-point scale (1 = not familiar at all, 5 = highly familiar) to ensure that these names were completely unacquainted with them.

96 pictures of hand activities such as chopping vegetables with a knife or doing a craft with scissors (see Figure 1) were selected from the previous study [47]. Each painful stimulus picture corresponds to a non-painful stimulus that depicts the same scene without the painful experience. The selected pictures were standardized to ensure consistent size (height 6.8 cm × width 8.4 cm), brightness and contrast.

Wong–Baker FACES Pain Rating Scale [48] was used in the pain intensity rating and the unpleasantness rating tasks (see Figure 2). The scale consists of six cartoon faces ranging from neutral to extreme pain, suitable for participants aged 3 and above. It is widely used in children’s pain assessment and research. A 6-point scoring system was employed. One point indicated no hurt or no unpleasantness, and 6 points represented the maximum pain or extreme unpleasantness.

#### 2.1.3. Procedure

The participants were led into a quiet room and signed an informed consent form before the experiment. Participants were told that the computer screen would display the name of a child they recognized or an unfamiliar child’s name, along with a picture illustrating a hand activity. In Experiment 1, participants completed both the pain intensity rating and the unpleasantness rating tasks, with each task consisting of 4 blocks, each containing 40 trials, preceded by a practice task of 10 trials. E-prime 3.0 software was used to present stimuli on a computer screen, starting with a fixation point presented for 300–500 ms randomly, followed by a blank screen for 500 ms. Subsequently, the names of in-group and out-group members, painful or non-painful pictures, and the FACES scale were presented simultaneously (Figure 3). Participants were instructed to press the keyboard keys 1–6 to indicate their pain intensity or unpleasantness rating scores, followed by a 500–1000 ms blank screen as an inter-trial interval. After the experiment, participants watched a 2 min video clip from “Tom and Jerry”. Their levels of unpleasantness were evaluated using the FACES scale before and after the video to ensure any potential negative impact on children was minimized (see Figure 3).

### 2.2. Results

Statistical analyses were performed using SPSS 26.0. Two within-participant factors—stimulus type (pain and non-painful) and peer relationship type (popular, rejected, neglected and unfamiliar)—were included for two-way repeated measures analysis of variance (ANOVA).

The ANOVA for pain intensity ratings revealed a significant main effect for stimulus type (*F* (1, 38) = 282.39, *p* < 0.001, *η_p_*^2^ = 0.88). Painful pictures (*M* = 4.24, *SE* = 0.13) received significantly higher ratings than non-painful pictures (*M* = 1.54, *SE* = 0.11, *p* < 0.001). The main effect of relationship type was also significant (*F* (3, 114) = 10.86, *p* < 0.001, *η_p_*^2^ = 0.22). The interaction between stimulus and peer relationship type was significant (*F* (3, 114) = 3.53, *p* = 0.017, *η_p_*^2^ = 0.09). Further analysis revealed that pain ratings for popular peers in painful conditions (*M* = 4.49, *SE* = 0.14) were significantly higher than those for rejected peers (*M* = 4.19, *SE* = 0.15, *p* = 0.011), neglected peers (*M* = 4.07, *SE* = 0.14, *p* < 0.001) and unfamiliar peers (*M* = 4.21, *SE* = 0.15, *p* = 0.011). Pain ratings for unfamiliar peers (*M* = 4.21, *SE* = 0.15) were significantly higher than those for neglected peers (*M* = 4.07, *SE* = 0.14, *p* = 0.041).

The results for unpleasantness ratings also showed a significant main effect of stimulus type (*F* (1, 38) = 157.84, *p* < 0.001, *η_p_*^2^ = 0.81). Unpleasantness ratings for painful pictures (*M* = 4.04, *SE* = 0.18) are significantly higher than those for non-painful pictures (*M* = 1.54, *SE* = 0.10, *p* < 0.001). The main effect of relationship type was significant (*F* (3, 114) = 4.01, *p* = 0.009, *η_p_*^2^ = 0.10). The interaction between stimulus and peer relationship type was significant (*F* (3, 114) = 3.38, *p* = 0.021, *η_p_*^2^ = 0.08). The unpleasantness ratings for popular peers in painful conditions (*M* = 4.21, *SE* = 0.20) were significantly higher than those for rejected peers (*M* = 3.97, *SE* = 0.17, *p* = 0.001), neglected peers (*M* = 4.01, *SE* = 0.17, *p* = 0.006) and unfamiliar peers (*M* = 3.96, *SE* = 0.21, *p* = 0.002). No other significant results were observed (see Figure 4).

### 2.3. Discussion

This study examined the influence of peer status on cognitive and affective empathy. The results showed that popular peers elicited higher pain and unpleasantness ratings, reflecting in-group favoritism in empathic processes. However, the results for the rejected and neglected peers did not demonstrate an advantage for in-group members in pain and unpleasantness ratings, with the ratings for neglected peers even being lower than those for unfamiliar peers. Previous research often uses the scores in pain and unpleasantness intensity ratings tasks as indicators of cognitive and affective empathy under experimental conditions [1]. Therefore, children in this study exhibited in-group favoritism in both cognitive and affective empathy, but the influence of peer status was more pronounced in cognitive empathy. Due to the limitations of behavioral experiments, experiment 2 will further explore the neural electrophysiological activities induced by pain in different peers using ERP technology.

## 3. Experiment 2

### 3.1. Methods

#### 3.1.1. Participants

28 children aged 8 to 12 years were randomly recruited for Experiment 2. All participants had normal or corrected vision and right-handedness, with no history of mental illness or brain injury, and were paid for their participation. Informed consent was obtained before the experiment. Due to excessive artifacts during EEG recordings, one participant was eliminated from the analyses. The final sample consisted of 27 participants (14 boys, 13 girls, *M* = 10.04 years, *SD* = 1.34 years). The participants in Experiment 2 were from different classes.

#### 3.1.2. Design and Material

The experiment also adopted the 2 stimulus type (pain stimulus, non-painful stimulus) × 4 peer relationship type (popular peer, rejected peer, neglected peer, unfamiliar peer) within-subject design. Painful and non-painful stimuli as well as pain intensity and subjective unpleasantness rating scale were taken from the sets of material used in Experiment 1.

Since the participants in this study were randomly recruited from the community, the peer nomination method used in Experiment 1 could not be applied uniformly. Therefore, before the experiment, participants were required to provide a list of classmates from their class and nominate 3 classmates they would most like to invite to their birthday party and 3 classmates they would least like to be grouped with during class activities. Same-gender classmates in the lists provided by participants who had not received any positive or negative nominations were randomly selected as neglected peers by researchers. Similar to Experiment 1, participants also assessed their familiarity with the names of unfamiliar peers using the 5-point scale (1 = not familiar at all, 5 = highly familiar).

#### 3.1.3. Procedure

Practice tasks consisting of 10 trials preceded the formal experiment. After a fixation point lasted for 400−600 ms randomly, participants were presented with the name of a peer for 1000 ms, followed by a blank screen for 500 ms and a hand activity picture for 1500 ms. Participants were instructed to imagine the peer doing a hand activity depicted in the picture and then complete the pain intensity rating task according to the prompt on the computer screen. There are 4 blocks in the whole experiment. Eighty trials were included in each block. The same measures as in Experiment 1 were taken after the experiment to eliminate any potential negative effects. Participants watched a 2 min video clip from “Tom and Jerry”, and their unpleasantness levels were assessed using the FACES scale (see Figure 5).

#### 3.1.4. EEG Recording

A 64-channel ERP system produced by Neuroscan, compliant with the international 10–20 system standard, was used for electrophysiological data collection. The electrode at the left mastoid was used as a reference and transformed to the average of the two mastoids offline. The band-pass filter was set to 0.05–100 Hz, and the sampling rate was 1000 Hz. The electrode impedance was kept at less than 5 kΩ.

#### 3.1.5. EEG Preprocessing

EEG data were preprocessed using the functions in Curry8 software (Version 8.0.5.0, Compumedics Neuroscan, Charlotte, NC, USA). First, the default reference electrode was set to the bilateral mastoid electrodes (M1 and M2). The missing electrodes were interpolated using spherical spline interpolation, where the sphere center of new electrodes is subtracted before the interpolation. It is a widely used method in EEG analysis [49]. Continuous EEG signals were band-pass filtered 0.1–30 Hz. Next, the detection range to 200 ms before and 200 ms after eye blinks was set to ocular artifact reduction. Epochs with amplitude values exceeding ±100 μV at any electrode were excluded from the average. In total, 16.17% of the trials were excluded due to artifacts (popular peer condition = 16.25%; rejected peer condition = 16.80%; neglected peer condition = 15.27%; unfamiliar peer condition = 16.34%). The data were segmented starting 200 ms before the onset of painful or non-painful pictures and lasting until 800 ms after these onsets. EEG epochs were base-line-corrected by a 200 ms time interval before the target stimuli onset. Finally, EEGLAB v2021.1 [50] was used to calculate the trial average for each participant and condition.

Based on topographic maps of the grand average and the results of previous studies [50,51], 4 ERP components were selected for analysis: N1 (120–160 ms), N2 (220–280 ms), P3 (300–400 ms) and LPP (450–800 ms). N1 and N2 amplitude were measured at frontocentral electrodes (F1, Fz, F2, FC1, FCz and FC2). P3 and LPP amplitude were measured at centroparietal electrodes (P3, Pz, P4, PO3, POz and PO4).

### 3.2. Results

#### 3.2.1. Behavioral Results

A 2 (stimulus type) × 4 (peer relationship type) repeated measures ANOVA was conducted, with pain intensity ratings as the dependent variable. There was a significant main effect for stimulus type (*F* (1, 26) = 184.90, *p* < 0.001, *η_p_*^2^ = 0.88). Pain stimuli received significantly higher ratings than non-painful stimuli (*p* < 0.001). There was also a significant main effect for peer relationship type (*F* (3, 78) = 4.38, *p* = 0.007, *η_p_*^2^ = 0.14). The interaction between stimulus type and relationship type was significant (*F* (3, 78) = 20.16, *p* < 0.001, *η_p_*^2^ = 0.44). In painful conditions, popular peers (*M* = 3.77, *SE* = 0.12) and unfamiliar peers (*M* = 3.47, *SE* = 0.10, *p* = 0.03) were rated higher than rejected peers (*M* = 3.30, *SE* = 0.10, *p* = 0.001), neglected peers (*M* = 3.05, *SE* = 0.14, *p* < 0.001). Unfamiliar peers (*M* = 3.47, *SE* = 0.10) were rated higher than rejected peers (*M* = 3.30, *SE* = 0.10, *p* = 0.008) and neglected peers (*M* = 3.05, *SE* = 0.14, *p* = 0.01) (see Figure 6).

#### 3.2.2. ERP Results

A 2 (stimulus type) × 4 (peer relationship type) repeated measures ANOVA was conducted on the peak amplitude of N1 and P3, as well as the mean amplitude of N2 and LPP.

N1 (120–160 ms)

For the N1 amplitude, there was not a significant main effect for stimulus type (*p* = 0.33) or for peer relationship type (*p* = 0.68). Additionally, the interactions between these two factors were not significant (*p* = 0.71) (see Figure 7).

N2 (220–280 ms)

No significant result was observed in the N2 amplitude. The main effects of stimulus type (*p* = 0.51), peer type (*p* = 0.62) and their interaction were not significant (*p* = 0.76) (see Figure 7).

P3 (300–400 ms)

For the P3 amplitude, there was a statistically significant main effect of stimulus type (*F* (1, 26) = 19.02, *p* < 0.001, *η_p_*^2^ = 0.42). In this context, painful pictures (*M* = 13.15, *SE* = 0.59) elicited larger P3 amplitude than non-painful pictures (*M* = 10.03, *SE* = 0.46). The main effect of peer relationship type was also significant (*F* (3, 78) = 2.97, *p* = 0.04, *η_p_*^2^ = 0.10). However, the interactions between these two factors were not significant (*p* = 0.17) (see Figure 8).

LPP (450–800 ms)

The main effect of stimulus type was insignificant (*F* (1, 26) = 27.49, *p* < 0.001, *η_p_*^2^ = 0.51). LPP amplitude was higher in painful conditions (*M* = 10.50, *SE* = 0.57) than in non-painful conditions (*M* = 6.96, *SE* = 0.46). The main effect of peer relationship type was insignificant (*F* (3, 78) = 3.98, *p* = 0.01, *η_p_*^2^ = 0.13). It also revealed a significant interaction between stimulus type and peer relationship type in the LPP amplitude (*F* (3, 78) = 2.84, *p* = 0.04, *η_p_*^2^ = 0.10). In painful conditions, LPP amplitude elicited by popular peers (*M* = 14.25, *SE* = 1.12) is larger than by rejected peers (*M* = 8.92, *SE* = 1.00, *p* = 0.004) and neglected peers (*M* = 7.63, *SE* = 0.80, *p* < 0.001). Moreover, unfamiliar peers (*M* = 11.18, *SE* = 1.43) elicited larger LPP amplitude than neglected peers (*M* = 7.63, *SE* = 0.80, *p* = 0.04) in painful conditions (see Figure 8).

### 3.3. Discussion

This study investigated the influence of peer status on neural activity in the early automatic and later controlled processes of pain empathy. The behavioral and electrophysiological results suggested the in-group favoritism in pain empathy. In painful conditions, popular peers elicited higher pain ratings. Unfamiliar peers elicited higher pain ratings than neglected peers and rejected peers. Additionally, in the ERP results, no significant findings were observed in the early components N1 and N2. However, we found the pain effect in empathic neural responses in the P3 and LPP components. The influence of peer status on children’s in-group favoritism in pain empathy was observed in the LPP component. Researchers suggested that the early components N1 and N2 reflect affective empathy, while the later components P3 and LPP indicate cognitive empathy [17,18,19]. Therefore, the results of both Experiment 1 and Experiment 2 showed that the influence of peer status on in-group favoritism within pain empathy in mid-childhood was concentrated in the cognitive evaluation processes.

## 4. General Discussion

Focusing on 8- to 12-year-old children, the present study investigated the influence of peer status on the in-group favoritism of children’s pain empathy in two experiments. In Experiment 1, we found that children in middle childhood exhibited significant in-group favoritism on pain and unpleasantness intensity rating tasks. In the pain intensity rating tasks, peers with lower status (especially neglected peers) elicited less cognitive empathy than unfamiliar peers. Experiment 2, which used ERP technology, yielded similar results. Differences between painful and non-painful conditions were observed in the late components P3 and LPP. Popular peers’ pain elicited the largest LPP amplitude. Besides, it is noteworthy that the pain of unfamiliar peers elicited larger LPP amplitudes than that of neglected peers.

The present study showed that children’s in-group favoritism during empathic processes is also affected by peer status. This study did not find in-group favoritism between unfamiliar peers and peers with neglected and rejected peers. On the contrary, the unfamiliar peers even elicited a greater pain empathy response, indicating that children’s peer status affects in-group favoritism in empathic processes. Researchers mainly focused on four interrelated but different aspects of peer status: social acceptance and rejection (social preference), social popularity and unpopularity (social impact). Social acceptance refers to the degree to which an individual is liked by peers, while social rejection reflects the degree to which an individual is disliked by peers [51]. Both Social acceptance and rejection show an individual’s interpersonal or affective status [52]. Social popularity and unpopularity represent an individual’s reputation within peer groups, describing the individual’s fame, power, reputation and prestige within peer groups [53]. Rejected children are more aggressive and dissocial. They have poorer social and cognitive abilities and may have higher levels of depression and anxiety than their average children. Coie [42] suggested that the behavior of socially anxious children may be more unpredictable and more likely to engage in embarrassing behaviors. Although researchers believe that the behavior of rejected children is likely to maintain or exacerbate social exclusion [43], the results of this study suggested that being ignored may cause more negative consequences during middle childhood than being rejected. Neglected children are less easily described due to their lower visibility within peer groups. Observational reports indicated that, compared to other children, neglected children often play alone, have poorer verbal communication skills [54], exhibit fewer aggressive interactions and show fewer positive social behaviors and traits than other children [43].

Although in-group favoritism can occur in any social group, most studies on in-group favoritism tend to examine obvious and formal groups, such as nations and races. In daily life, people often spontaneously form various groups during interpersonal communication. These groups are often not bounded by language or skin color and are difficult to distinguish by visual features, but they are widely present in daily life. Saarinen et al. [16] conducted a meta-analysis of 87 fMRI studies and found that different group types lead to different responses in the brain regions activated by group bias. The bias induced by trivial groups—those artificially created and differentiated—is more pronounced in the cingulate cortex, whereas intergroup bias resulting from real groups, such as racial, ethnic or political affiliations, involves a broader range of brain regions. In addition, there are no intergroup biases at the neural level in moral processing, social judgment or higher-order cognitive processes, such as memory and learning. However, significant neural intergroup biases are observed in facial processing and empathy-related cognitive pathways. In the present study, although there were no predefined rules for creating and differentiating groups, children still showed an in-group bias in empathy for pain when facing different peers.

Pain empathy could be evaluated using empathy questionnaires [55,56,57]. It could also be evaluated by subjective ratings and brain responses to visual stimuli depicting or indicating that another person is in pain [17,58,59]. In our study, both behavioral and electrophysiological results revealed significant differences between painful and non-painful pictures and peer status exerting a greater influence on children’s cognitive empathy processes than on their affective empathy processes. Early electroencephalographic components N1 and N2 are posited to reflect the early automated affective sharing in empathy for pain, that is, affective empathy. Late components P3 and LPP are thought to reflect the cognitive evaluation of painful stimuli, which pertains to cognitive empathy. According to the results of our study, in-group bias and peer status primarily affect the top-down sustained attention processing and cognitive evaluation of pain empathy, consistent with the results of Mathur et al. ’s [60] study. Mathur et al. [60] used fMRI to investigate racial group biases and discovered that the pain of in-group and out-group members led to differences in the medial prefrontal cortex, highlighting the role of cognitive empathy in in-group favoritism. The absence of differences in stimulus type and peer type in early electroencephalographic components may be attributed to the high heterogeneity of N1 and N2 components in pain empathy research. A meta-analysis indicated that the results of N1 and N2 are susceptible to various influences during the experimental processes (e.g., experimental paradigms, materials, etc.). In contrast, later electroencephalographic components P3 and LPP are considered robust indicators of pain empathy [61].

Moreover, the age of the participants may also be one of the factors affecting the research outcomes. Cheng et al. [62] conducted a study with children aged between 4.5 and 9 years and found age-related changes in the amplitude of N2 and LPP components. As age increased, the amplitude of N2 decreased while the amplitude of LPP increased, indicating a shift in children from personal affective sharing to actions such as helping and comforting. Numerous studies have shown that the LPP is associated with cognitive evaluation in typically developing children, and an increase in the LPP amplitude appears to be an effective indicator of the maturity of cognitive evaluation during childhood [63,64,65,66,67,68,69]. Cognitive empathy, which relies more on the development of the prefrontal cortex, is more significant in mid-childhood individuals due to the relatively advanced development of the frontal lobe. Therefore, during mid-childhood, the proportion of cognitive empathy is greater than that of affective empathy [5].

There are a few limitations in the present study. Firstly, the sample comprised only children with an average age of 9 to 10 years and did not encompass a broader age spectrum. Therefore, the present study could not explore the potential effects of age development, which to some extent limits the generalizability of the results. Future research should consider incorporating a broader age range to enhance the applicability of the findings. Secondly, the peer nomination method used to select peers of children was not uniform between Experiment 1 and Experiment 2, which may have exerted a certain influence on the experimental results. Besides, the use of subjective reports may introduce biases in the process of determining different peers. Future research should consider incorporating more objective measures. At last, this study used the presentation of peer names to prompt participants to imagine others’ experiences. Future studies could enhance the vividness of the experimental conditions for participants by substituting names with photographs of peers.

## 5. Conclusions

Children in middle childhood demonstrate in-group favoritism during the cognitive evaluation process of pain empathy, while this phenomenon is not observed in the emotional sharing process. Furthermore, children’s in-group favoritism is influenced by peer status: popular peers induce the strongest levels of pain empathy, and unfamiliar peers elicit greater pain empathy than peers who are rejected or neglected.

## Figures and Tables

**Figure 1 brainsci-14-01262-f001:**
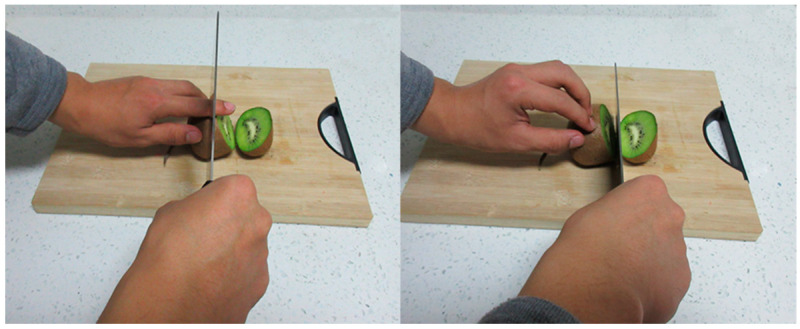
Examples of painful (left column) and non-painful (right column) pictures.

**Figure 2 brainsci-14-01262-f002:**
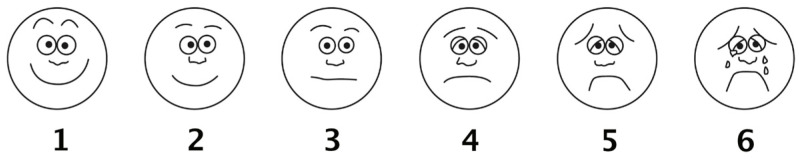
Wong–Baker FACES Pain Rating Scale.

**Figure 3 brainsci-14-01262-f003:**
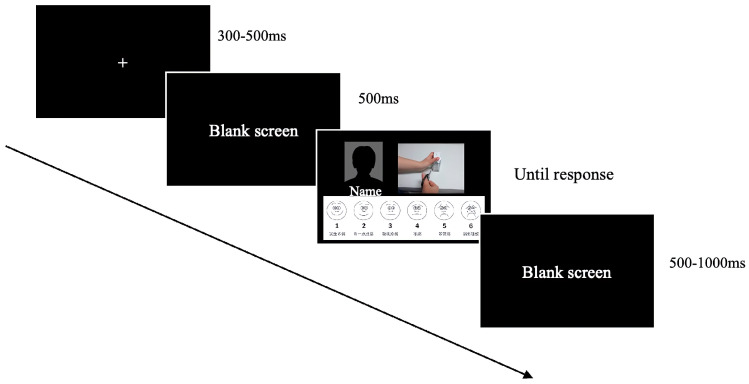
Procedure of Experiment 1.

**Figure 4 brainsci-14-01262-f004:**
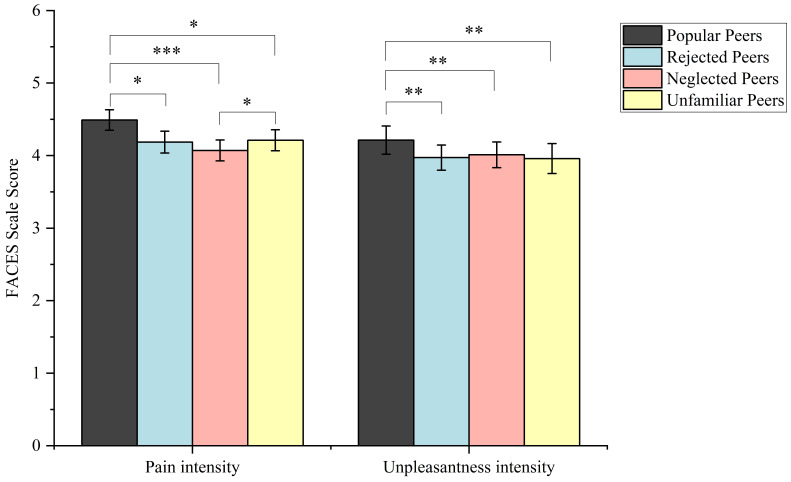
FACES scale rating scores by different tasks in painful conditions. Error bars represent standard errors. * *p* < 0.05 ** *p* < 0.01 *** *p* < 0.001.

**Figure 5 brainsci-14-01262-f005:**
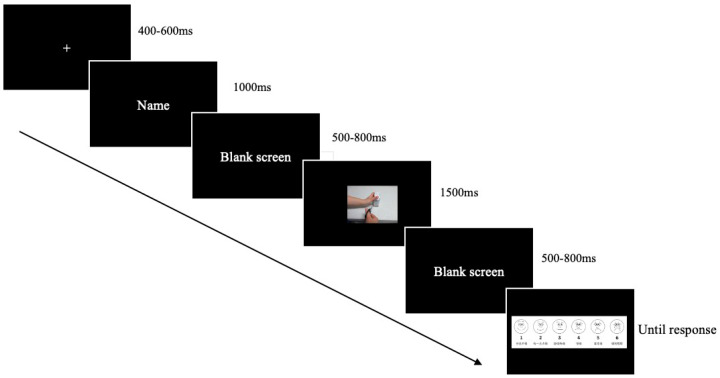
Procedure of Experiment 2.

**Figure 6 brainsci-14-01262-f006:**
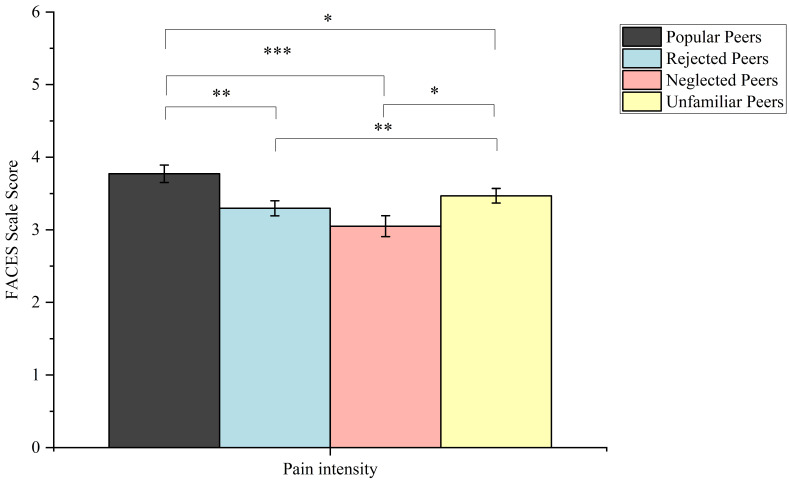
FACES scale rating scores by pain intensity rating tasks in painful conditions. Error bars represent standard errors. * *p* < 0.05 ** *p* < 0.01 *** *p* < 0.001.

**Figure 7 brainsci-14-01262-f007:**
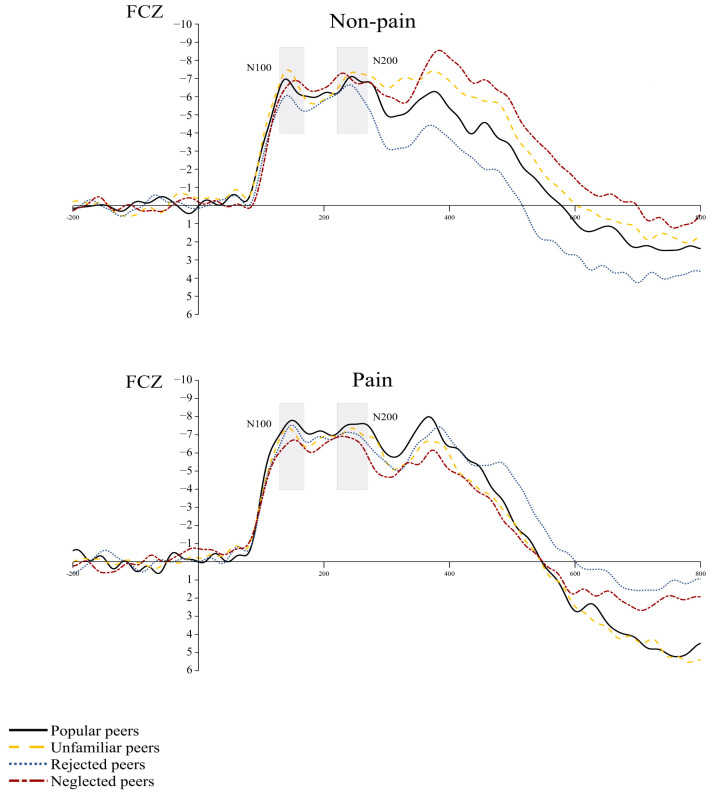
Illustration of ERPs at electrodes FCZ in response to painful and neutral stimulus in Experiment 2. The gray areas indicated the time windows for N1 and N2 amplitudes.

**Figure 8 brainsci-14-01262-f008:**
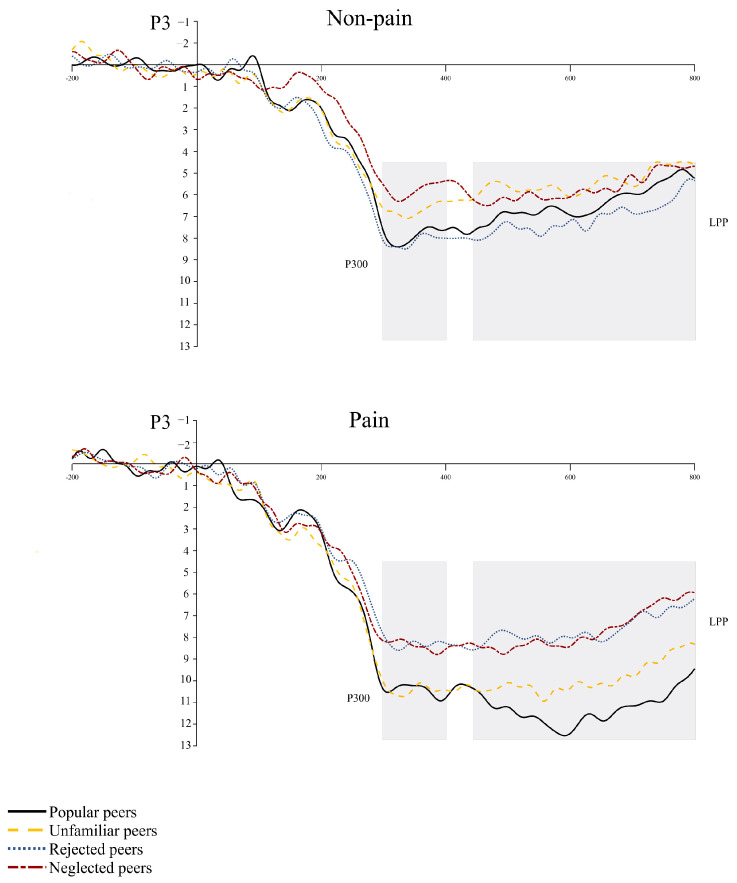
Illustration of ERPs at electrodes P3 in response to painful and neutral stimulus in Experiment 2. The gray areas indicated the time windows for P3 and LPP amplitudes.

## Data Availability

Due to the need to safeguard the privacy of the minor participants in our study, the data underlying this article cannot be publicly shared. This includes sensitive information such as the minors’ names. However, to uphold the principles of research transparency and reproducibility, we are willing to share the data upon reasonable request to the corresponding author. The raw data supporting the conclusions of this article will be made available by the authors on request, ensuring that the privacy of our young participants is maintained while facilitating scholarly inquiry.

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
