# Peer review of "Peer Status Influences In-Group Favoritism in Pain Empathy During Middle Childhood: Evidence from Behavioral and Event-Related Potentials Studies"

_brainsci, 2024, doi:10.3390/brainsci14121262_

Round 1
Reviewer 1 Report
Comments and Suggestions for Authors
This study investigated how peer status influences in-group favoritism in pain empathy. The findings concluded that children in middle childhood exhibited in-group favoritism during the cognitive evaluation of pain empathy, but not during the emotional sharing process. Peer status significantly affects this favoritism. The idea is creative, and the paper is well-written in structure. However, technical details are lacking, and some critical issues need to be addressed.
Provide references for the tools used, such as Curry8 and EEGLAB. Additionally, clarify which tool was used for each preprocessing step. Without this information, replication of the study becomes challenging.
The specifications of the band-pass filter used were not revealed. Please provide details, including filter type and all the relevant parameters, with how you did (e.g., you might use EEGLAB function or start with some basic functions in MATLAB).
EEG signals are highly sensitive, and a band-pass filter alone cannot address all potential noise sources. The lack of detailed preprocessing raises concerns about the validity of the recordings. If additional preprocessing steps were conducted, such as ICA or ASR, describe these in full, including the parameters used. Common preprocessing steps usually involve typical steps like channel rejection, interpolation, rereferencing, and trial rejection. Without these details, it is difficult to evaluate the quality of the data and the reliability of the results. I strongly recommend creating a dedicated subsection titled "EEG Preprocessing" to ensure that all steps and parameters are transparently documented. This will enhance the study's replicability and credibility.
Plus, the authors did not disclose how many trials were discarded. This is a critical omission, as the number of discarded trials directly impacts the analysis and interpretability of the findings. Please provide these details.
The definitions of the peer relationship groups (particularly popular, rejected, and neglected) are subjective and may introduce biases, both from the children providing the nominations and from the researchers interpreting the data. While subjective reports are often acceptable for emotional ratings, they are more problematic for determining group classifications like peer status. Consider incorporating more objective measures.
Regarding the subjective measures, I think relying solely on raw scores ignores individual response tendencies and can weaken the robustness of the findings. The authors should address the response bias issue which can distort results. Otherwise, comparisons across participants might reflect differences in rating styles rather than genuine differences in perceived empathy or recognition.
Author Response
Dear reviewer, thank you very much for finding interest in our findings and pointing out the flaws in our manufacture. We have addressed your concerns in a point-by-point manner below, and hope that you will find the added information suitable and sufficient for publication.
Comments 1: Provide references for the tools used, such as Curry8 and EEGLAB. Additionally, clarify which tool was used for each preprocessing step. Without this information, replication of the study becomes challenging.
Response 1: We sincerely thank you for your careful reading and pointing this out. We agree with this comment. We have added this information in EEG preprocessing in the Method of Experiment 2: EEG data were preprocessed using the functions in Curry8 software (Compumedics Neuroscan, Charlotte, NC, USA). First, the default reference electrode was set to the bilateral mastoid electrodes (M1 and M2). The missing electrodes were interpolated. Continuous EEG signals were band-pass filtered 0.1–30 Hz. Next, the detection range to 200 ms before and 200 ms after eye blinks was set to ocular artifact reduction. Epochs with amplitude values exceeding ± 100 μV at any electrode were excluded from the average. 16.17% of the trials were excluded due to artifacts (popular peer condition = 16.25%; rejected peer condition = 16.80%; neglected peer condition = 15.27%; unfamiliar peer condition = 16.34%). The data were segmented starting 200 ms before the onset of painful or nonpainful pictures and lasting until 800 ms after these onsets. EEG epochs were base-line-corrected by a 200 ms time interval before the target stimuli onset. Finally, EEGLAB v2021.1 (Delorme & Makeig, 2004) was used to calculate the trial average for each participant and condition. (Page 8, Paragraph 4, Lines 341−352)
References
Delorme, A.; Makeig, S. EEGLAB: An Open Source Toolbox for Analysis of Single-Trial EEG Dynamics Including Independent Component Analysis. Journal of Neuroscience Methods 2004, 134, 9–21.
Comments 2: The specifications of the band-pass filter used were not revealed. Please provide details, including filter type and all the relevant parameters, with how you did (e.g., you might use EEGLAB function or start with some basic functions in MATLAB).
Response 2: We think this is an excellent suggestion. To make the content of the article clearer, we have added details about the EEG data recording and preprocessing:
- A 64-channel ERP system produced by Neuroscan, compliant with the international 10–20 system standard, was used for electrophysiological data collection. The electrode at the left mastoid was used as a reference and transformed to the average of the two mastoids offline. The band-pass filter was set to 0.05–100Hz, and the sampling rate was 1000 Hz. The electrode impedance was kept at less than 5 kΩ. (Page 8, Paragraph 3, Lines 334−339)
- EEG data were preprocessed using the functions in Curry8 software (Compumedics Neuroscan, Charlotte, NC, USA). First, the default reference electrode was set to the bilateral mastoid electrodes (M1 and M2). The missing electrodes were interpolated. Continuous EEG signals were band-pass filtered 0.1–30 Hz. Next, the detection range to 200 ms before and 200 ms after eye blinks was set to ocular artifact reduction. Epochs with amplitude values exceeding ± 100 μV at any electrode were excluded from the average. 16.17% of the trials were excluded due to artifacts (popular peer condition = 16.25%; rejected peer condition = 16.80%; neglected peer condition = 15.27%; unfamiliar peer condition = 16.34%). The data were segmented starting 200 ms before the onset of painful or nonpainful pictures and lasting until 800 ms after these onsets. EEG epochs were base-line-corrected by a 200 ms time interval before the target stimuli onset. Finally, EEGLAB v2021.1 (Delorme & Makeig, 2004) was used to calculate the trial average for each participant and condition. (Page 8, Paragraph 4, Lines 341−352)
Comments 3: EEG signals are highly sensitive, and a band-pass filter alone cannot address all potential noise sources. The lack of detailed preprocessing raises concerns about the validity of the recordings. If additional preprocessing steps were conducted, such as ICA or ASR, describe these in full, including the parameters used. Common preprocessing steps usually involve typical steps like channel rejection, interpolation, rereferencing, and trial rejection. Without these details, it is difficult to evaluate the quality of the data and the reliability of the results. I strongly recommend creating a dedicated subsection titled "EEG Preprocessing" to ensure that all steps and parameters are transparently documented. This will enhance the study's replicability and credibility.
Response 3: We are grateful for the suggestion. We selected the target brain regions and electrodes of interest based on previous research (Han, 2018; Kiat & Cheadle, 2017): N1 and N2 amplitudes were measured at frontocentral electrodes (F1, Fz, F2, FC1, FCz, and FC2), while P3 and LPP amplitudes were measured at centroparietal electrodes (P3, Pz, P4, PO3, POz, and PO4). During data acquisition, the signal quality of these electrodes was good, and channel rejection was not required. Our preprocessing steps did not include ICA or ASR, but artifacts such as ocular signals were removed following methods used in previous studies to ensure the purity of the EEG data (Zhang, 2024; Flanagan et al., 2020). According to your advice, we have added “EEG Preprocessing” in the Methods of Experiment 2 and the steps of ERP data preprocessing was described in detail: EEG data were preprocessed using the functions in Curry8 software (Compumedics Neuroscan, Charlotte, NC, USA). First, the default reference electrode was set to the bilateral mastoid electrodes (M1 and M2). The missing electrodes were interpolated. Continuous EEG signals were band-pass filtered 0.1–30 Hz. Next, the detection range to 200 ms before and 200 ms after eye blinks was set to ocular artifact reduction. Epochs with amplitude values exceeding ± 100 μV at any electrode were excluded from the average. 16.17% of the trials were excluded due to artifacts (popular peer condition = 16.25%; rejected peer condition = 16.80%; neglected peer condition = 15.27%; unfamiliar peer condition = 16.34%). The data were segmented starting 200 ms before the onset of painful or nonpainful pictures and lasting until 800 ms after these onsets. EEG epochs were base-line-corrected by a 200 ms time interval before the target stimuli onset. Finally, EEGLAB v2021.1 (Delorme & Makeig, 2004) was used to calculate the trial average for each participant and condition. (Page 8, Paragraph 4, Lines 341−352)
References
Flanagan, S. D., Proessl, F., Dunn-Lewis, C., Canino, M. C., Sterczala, A. J., Connaboy, C., DuPont, W. H., Caldwell, L. K., & Kraemer, W. J. (2020). Constitutive and Stress-Induced Psychomotor Cortical Responses to Compound K Supplementation. Frontiers in Neuroscience, 14.
Han S. (2018). Neurocognitive Basis of Racial Ingroup Bias in Empathy. Trends in cognitive sciences, 22(5), 400–421.
Kiat, J. E., & Cheadle, J. E. (2017). The impact of individuation on the bases of human empathic responding. NeuroImage, 155, 312–321.
Zhang, P., Cao, L., Yuan, J., Wang, C., Ou, Y., Wang, J., Duan, L., Qian, H., Ling, Q., & Yuan, X. (2024). Early Impairment of Face Perception in Post-Stroke Depression: An ERP Study. Clinical EEG and Neuroscience, 15500594241289473.
Comments 4: Plus, the authors did not disclose how many trials were discarded. This is a critical omission, as the number of discarded trials directly impacts the analysis and interpretability of the findings. Please provide these details.
Response 4: Thank you for continuing to point out the details. In the newest manuscript, we have explained how many trials were discarded: EEG data were preprocessed using the functions in Curry8 software (Compumedics Neuroscan, Charlotte, NC, USA). First, the default reference electrode was set to the bilateral mastoid electrodes (M1 and M2). The missing electrodes were interpolated. Continuous EEG signals were band-pass filtered 0.1–30 Hz. Next, the detection range to 200 ms before and 200 ms after eye blinks was set to ocular artifact reduction. Epochs with amplitude values exceeding ± 100 μV at any electrode were excluded from the average. 16.17% of the trials were excluded due to artifacts (popular peer condition = 16.25%; rejected peer condition = 16.80%; neglected peer condition = 15.27%; unfamiliar peer condition = 16.34%). The data were segmented starting 200 ms before the onset of painful or nonpainful pictures and lasting until 800 ms after these onsets. EEG epochs were base-line-corrected by a 200 ms time interval before the target stimuli onset. Finally, EEGLAB v2021.1 (Delorme & Makeig, 2004) was used to calculate the trial average for each participant and condition. (Page 8, Paragraph 4, Lines 341−352)
Comments 5: The definitions of the peer relationship groups (particularly popular, rejected, and neglected) are subjective and may introduce biases, both from the children providing the nominations and from the researchers interpreting the data. While subjective reports are often acceptable for emotional ratings, they are more problematic for determining group classifications like peer status. Consider incorporating more objective measures.
Response 5: We are grateful for the suggestions. In Experiment 1, we employed the peer nomination method, which is widely used in research on children's peer status. However, in Experiment 2, it was challenging to conduct peer nominations within the classroom and subsequently bring child participants to the EEG laboratory to collect EEG data. Therefore, after randomly recruiting participants, we asked them to nominate their most liked and least liked same-gender classmates based on a list of classmates they provided. Finally, the researcher randomly selected a same-gender peer who had not been nominated to serve as the neglected peer. This procedure might have impacted the objectivity of the results, which we addressed at the end of the Discussion section: Secondly, the peer nomination method used to select peers of children was not uniform between Experiment 1 and Experiment 2, which may have exerted a certain influence on the experimental results. Besides, the use of subjective reports may introduce biases in the process of determining different peers. Future research should consider incorporating more objective measures. (Page 12, Paragraph 4, Lines 505−509)
Comments 6: Regarding the subjective measures, I think relying solely on raw scores ignores individual response tendencies and can weaken the robustness of the findings. The authors should address the response bias issue which can distort results. Otherwise, comparisons across participants might reflect differences in rating styles rather than genuine differences in perceived empathy or recognition.
Response 6: We gratefully appreciate your valuable comment. Through an extensive review of the literature, we found that pain empathy could be evaluated using empathy questionnaires (Fusaro, Tieri, & Aglioti, 2016; Nazarewicz, Verdejo-Garcia, & Giummarra, 2015; Zhao et al., 2020). It could also be evaluated by subjective ratings and brain responses to visual stimuli depicting or indicating that another person is in pain (Cui, Zhu, & Luo, 2017; Decety, Yang, & Cheng, 2010; Fan & Han, 2008). Subjective measures are frequently used in research on pain empathy to assess participants' perceived pain intensity and the unpleasantness of the stimuli. The statistical analysis results of the raw scores from subjective measures are often presented in the Behavioral Results section (Ren et al., 2019; Wu et al., 2020; Meng et al., 2013; Yan et al., 2017). Besides, the painful and non-painful pictures used in our study were pre-evaluated, and there were significant differences between the painful and non-painful pictures. Moreover, participant rating style differences may exist, but our within-subject experimental design may help mitigate potential interindividual differences. We focused on the differences in pain ratings across different peer-type conditions for all participants.
References
Cui, F., Zhu, X., & Luo, Y. (2017). Social contexts modulate neural responses in the processing of others’ pain: An event-related potential study. Cognitive, Affective, & Behavioral Neuroscience, 17(4), 850–857.
Decety, J., Yang, C. Y., & Cheng, Y. (2010). Physicians down-regulate their pain empathy response: An event-related brain potential study. NeuroImage, 50(4), 1676–1682.
Fan, Y., & Han, S. (2008). Temporal dynamic of neural mechanisms involved in empathy for pain: An event-related brain potential study. Neuropsychologia, 46(1), 160–173.
Fusaro, M., Tieri, G., & Aglioti, S. M. (2016). Seeing pain and pleasure on self and others: Behavioral and psychophysiological reactivity in immersive virtual reality. Journal of Neurophysiology, 116(6), 2656–2662.
Meng, J., Jackson, T., Chen, H., Hu, L., Yang, Z., Su, Y., & Huang, X. (2013). Pain perception in the self and observation of others: an ERP investigation. NeuroImage, 72, 164–173.
Nazarewicz, J., Verdejo-Garcia, A., & Giummarra, M. J. (2015). Sympathetic pain? A role of poor parasympathetic nervous system engagement in vicarious pain states. Psychophysiology, 52(11), 1529–1537.
Ren, Q., Lu, X., Zhao, Q., Zhang, H., & Hu, L. (2020). Can self-pain sensitivity quantify empathy for others' pain?. Psychophysiology, 57(10), e13637.
Wu, Y. J., Liu, Y., Yao, M., Li, X., & Peng, W. (2020). Language contexts modulate instant empathic responses to others' pain. Psychophysiology, 57(8), e13562.
Yan, Z., Pei, M., & Su, Y. (2017). Children's Empathy and Their Perception and Evaluation of Facial Pain Expression: An Eye Tracking Study. Frontiers in psychology, 8, 2284.
Zhao, Q., Ren, Q., Sun, Y., Wan, L., & Hu, L. (2020). Impact factors of empathy in mainland Chinese youth. Frontiers in Psychology, 11, 688.
We sincerely appreciate the time and effort the reviewers and editors have invested in providing insightful and constructive feedback on our manuscript. These suggestions have been invaluable in improving the quality and clarity of our work. We have carefully addressed all comments and implemented the recommended changes to the best of our ability. Should there be any remaining concerns or areas requiring further clarification, we would be more than happy to revise the manuscript further. Thank you again for your thoughtful guidance and support throughout this process.

Reviewer 2 Report
Comments and Suggestions for Authors
Stating that "In-groups are the groups with which individuals primarily interact" could give a wrong impression, because in-groups are those to which the individual belongs, but the individual could interact with some out-groups, too.
When you describe "According to a previous study [37], social preference (positive nominate scores minus negative nominate scores) and social impact scores (positive nominate scores plus negative nominate scores) were calculated to classify the children into different peer groups." it is better if you specify which these groups are - later in the Results section, one can read about popular peers, neglected peers, rejected peers, familiar peers, unfamiliar peers. It is better if you specify how each group is determined to become clear if only familiar peers could be popular peers, neglected peers, and rejected peers.
Describing Experiment 1 and Experiment 2, please, specify if all participants were from the same class at school. You state "Same-gender classmates who had not received any positive or negative nominations were randomly selected as neglected peers." How do you know the names of non-selected classmates?
It is better if you explain the abbreviation ERP - Event-Related Potentials, when you first use it in the article.
Author Response
Dear reviewer, thank you very much for finding interest in our findings and pointing out the flaws in our manufacture. We have addressed your concerns in a point-by-point manner below, and hope that you will find the added information suitable and sufficient for publication.
Comments 1: Stating that "In-groups are the groups with which individuals primarily interact" could give a wrong impression, because in-groups are those to which the individual belongs, but the individual could interact with some out-groups, too.
Response 1: We sincerely thank you for your careful reading and pointing this out. We agree with this comment. This sentence indeed fails to accurately convey the concept of in-groups. Therefore, we have deleted this sentence in the revised manuscript (Page 1, Paragraph 2, Line 39).
Comments 2: When you describe "According to a previous study [37], social preference (positive nominate scores minus negative nominate scores) and social impact scores (positive nominate scores plus negative nominate scores) were calculated to classify the children into different peer groups." it is better if you specify which these groups are - later in the Results section, one can read about popular peers, neglected peers, rejected peers, familiar peers, unfamiliar peers. It is better if you specify how each group is determined to become clear if only familiar peers could be popular peers, neglected peers, and rejected peers.
Response 2: We think this is an excellent suggestion. To make the content of the article clearer, we have added details about the peer nomination method, including the formulas for distinguishing different types of peers: Positive and negative nomination counts were obtained and standardized within the class, and Z-scores were calculated. All children in the class got a positive nomination score (PN) and a standardized negative nomination score (NN). Besides, social preference (SP, positive nominate scores minus negative nominate scores) and social impact scores (SI, positive nominate scores plus negative nominate scores) were also calculated. The previous study [37] has developed a formula to classify children into sociometric groups: popular (SP > 1, PN > 0, NN < 0), rejected (SP < −1, PN < 0, NN > 0), neglected (SI < −1, PN < 0, NN < 0), controversial (SI > 1, PN > 0, NN > 0) and average (Not in previous categories) (Page 5, Paragraph 2, Lines 208−215).
Comments 3: Describing Experiment 1 and Experiment 2, please, specify if all participants were from the same class at school. You state "Same-gender classmates who had not received any positive or negative nominations were randomly selected as neglected peers." How do you know the names of non-selected classmates?
Response 3: We are grateful for the suggestion. According to your advice, we have added more interpretations to the Methods of Experiment 1 and Experiment 2:
- 40 fourth-grade elementary school students (20 boys, 20 girls) from four classes were enrolled for Experiment 1. (Page 4, Paragraph 3, Lines 194−195)
- Before the formal experiment, a limited nomination method was used in the four classes from which the subjects originated. (Page 4, Paragraph 2, Lines 205−206)
- The participants in Experiment 2 were from different classes. (Page 7, Paragraph 2, Lines 304−305)
- Same-gender classmates in the lists provided by participants who had not received any positive or negative nominations were randomly selected as neglected peers by researchers. (Page 8, Paragraph 1, Lines 316−317)
Comments 4: It is better if you explain the abbreviation ERP - Event-Related Potentials, when you first use it in the article.
Response 4: Thank you for continuing to point out the details. We have explained the ERP in the Abstract (Page 1, Lines 13−14) and Introduction (Page 2, Paragraph 2, Line 60).
We sincerely appreciate the time and effort the reviewers and editors have invested in providing insightful and constructive feedback on our manuscript. These suggestions have been invaluable in improving the quality and clarity of our work. We have carefully addressed all comments and implemented the recommended changes to the best of our ability. Should there be any remaining concerns or areas requiring further clarification, we would be more than happy to revise the manuscript further. Thank you again for your thoughtful guidance and support throughout this process.

Round 2
Reviewer 1 Report
Comments and Suggestions for Authors
I read through the authors’ response. However, I found some issues have not been addressed well.
Comments 2: The specifications of the band-pass filter used were not revealed. Please provide details, including filter type and all the relevant parameters, with how you did (e.g., you might use EEGLAB function or start with some basic functions in MATLAB).
Response 2: We think this is an excellent suggestion. To make the content of the article clearer, we have added details about the EEG data recording and preprocessing:
[…]
ð I found nothing have been added for this issue. The cutoff frequencies and sampling rate were already mentioned in the first version of the paper. Please provide the spec of the filter, including filter types, filter order, used window, etc.. with the tool or a specific function for computation. You should reveal all the relevant parameters.
Comments 3: EEG signals are highly sensitive, and a band-pass filter alone cannot address all potential noise sources. The lack of detailed preprocessing raises concerns about the validity of the recordings. If additional preprocessing steps were conducted, such as ICA or ASR, describe these in full, including the parameters used. Common preprocessing steps usually involve typical steps like channel rejection, interpolation, rereferencing, and trial rejection. Without these details, it is difficult to evaluate the quality of the data and the reliability of the results. I strongly recommend creating a dedicated subsection titled "EEG Preprocessing" to ensure that all steps and parameters are transparently documented. This will enhance the study's replicability and credibility.
Response 3: We are grateful for the suggestion. We selected the target brain regions and electrodes of interest based on previous research (Han, 2018; Kiat & Cheadle, 2017): N1 and N2 amplitudes were measured at frontocentral electrodes (F1, Fz, F2, FC1, FCz, and FC2), while P3 and LPP amplitudes were measured at centroparietal electrodes (P3, Pz, P4, PO3, POz, and PO4). During data acquisition, the signal quality of these electrodes was good, and channel rejection was not required. Our preprocessing steps did not include ICA or ASR, but artifacts such as ocular signals were removed following methods used in previous studies to ensure the purity of the EEG data (Zhang, 2024; Flanagan et al., 2020). According to your advice, we have added “EEG Preprocessing” in the Methods of Experiment 2 and the steps of ERP data preprocessing was described in detail: EEG data were preprocessed using the functions in Curry8 software (Compumedics Neuroscan, Charlotte, NC, USA). First, the default reference electrode was set to the bilateral mastoid electrodes (M1 and M2). The missing electrodes were interpolated. Continuous EEG signals were band-pass filtered 0.1–30 Hz. Next, the detection range to 200 ms before and 200 ms after eye blinks was set to ocular artifact reduction. Epochs with amplitude values exceeding ± 100 μV at any electrode were excluded from the average. 16.17% of the trials were excluded due to artifacts (popular peer condition = 16.25%; rejected peer condition = 16.80%; neglected peer condition = 15.27%; unfamiliar peer condition = 16.34%). The data were segmented starting 200 ms before the onset of painful or nonpainful pictures and lasting until 800 ms after these onsets. EEG epochs were base-line-corrected by a 200 ms time interval before the target stimuli onset. Finally, EEGLAB v2021.1 (Delorme & Makeig, 2004) was used to calculate the trial average for each participant and condition. (Page 8, Paragraph 4, Lines 341−352)
ð Explicitly mention the method used for channel interpolation. It is likely that spherical spline interpolation was used, as it is a standard option in CURRY8 software or the default option in EEGLAB, and is widely utilized in EEG studies [1]. You could include a statement like: 'The missing electrodes were interpolated using spherical spline interpolation, which is a widely used method in EEG analysis [citation].' This would provide a brief justification of using the method and ensure the replicability.
[1] https://doi.org/10.3389/frsip.2023.1064138
Comments 6: Regarding the subjective measures, I think relying solely on raw scores ignores individual response tendencies and can weaken the robustness of the findings. The authors should address the response bias issue which can distort results. Otherwise, comparisons across participants might reflect differences in rating styles rather than genuine differences in perceived empathy or recognition.
Response 6: We gratefully appreciate your valuable comment. Through an extensive review of the literature, we found that pain empathy could be evaluated using empathy questionnaires (Fusaro, Tieri, & Aglioti, 2016; Nazarewicz, Verdejo-Garcia, & Giummarra, 2015; Zhao et al., 2020). It could also be evaluated by subjective ratings and brain responses to visual stimuli depicting or indicating that another person is in pain (Cui, Zhu, & Luo, 2017; Decety, Yang, & Cheng, 2010; Fan & Han, 2008). Subjective measures are frequently used in research on pain empathy to assess participants' perceived pain intensity and the unpleasantness of the stimuli. The statistical analysis results of the raw scores from subjective measures are often presented in the Behavioral Results section (Ren et al., 2019; Wu et al., 2020; Meng et al., 2013; Yan et al., 2017). Besides, the painful and non-painful pictures used in our study were pre-evaluated, and there were significant differences between the painful and non-painful pictures. Moreover, participant rating style differences may exist, but our within-subject experimental design may help mitigate potential interindividual differences. We focused on the differences in pain ratings across different peer-type conditions for all participants.
ð These are good points. Please incorporate them into the discussion.
Author Response
Dear reviewer, thank you for your valuable suggestions. In response to your suggestions, we have made corresponding revisions to the manuscript. We have addressed your concerns in a point-by-point manner below, and hope that you will find the added information suitable and sufficient for publication.
Comment 2: The specifications of the band-pass filter used were not revealed. Please provide details, including filter type and all the relevant parameters, with how you did (e.g., you might use EEGLAB function or start with some basic functions in MATLAB).
ð I found nothing have been added for this issue. The cutoff frequencies and sampling rate were already mentioned in the first version of the paper. Please provide the spec of the filter, including filter types, filter order, used window, etc.. with the tool or a specific function for computation. You should reveal all the relevant parameters.
Response 2: Thank you for your kind suggestions. We have reviewed the EEG preprocessing process and consulted the Curry8 software manual. The Curry8 software offers three types of filters: Bandpass Filter, Notch Filter, and Bandstop Filter. The Bandpass Filter is the most commonly used and the most suitable for this study. During the data collection process, the 50 Hz mains interference was already avoided, so the Notch Filter is not required. The Bandstop Filter is primarily used to suppress signals within a specific frequency range while allowing frequencies outside that range to pass, which is the opposite of the Bandpass Filter's functionality. In the Bandpass Filter of Curry8, the User Defined (Auto) option automatically sets the slopes. The preset bandpass filters available include Delta-Band, Theta-Band, Alpha-Band, and others. The Ripples and Fast Ripples options are designed to select higher-frequency bands to focus on High-Frequency Oscillations (HFO) associated with epilepsy. Ripples have a frequency range of 80–200 Hz, and Fast Ripples cover a range of 200–450 Hz.
For our study, we selected the User Defined (Auto) option for filtering and set the High Pass and Low Pass parameters to 0.1 Hz and 30 Hz. In addition, no further filtering operations are required in the Curry8 software. Therefore, we only reported the high-pass and low-pass filter parameters of the Bandpass filter in the article.
Comments 3: EEG signals are highly sensitive, and a band-pass filter alone cannot address all potential noise sources. The lack of detailed preprocessing raises concerns about the validity of the recordings. If additional preprocessing steps were conducted, such as ICA or ASR, describe these in full, including the parameters used. Common preprocessing steps usually involve typical steps like channel rejection, interpolation, rereferencing, and trial rejection. Without these details, it is difficult to evaluate the quality of the data and the reliability of the results. I strongly recommend creating a dedicated subsection titled "EEG Preprocessing" to ensure that all steps and parameters are transparently documented. This will enhance the study's replicability and credibility.
ð Explicitly mention the method used for channel interpolation. It is likely that spherical spline interpolation was used, as it is a standard option in CURRY8 software or the default option in EEGLAB, and is widely utilized in EEG studies [1]. You could include a statement like: 'The missing electrodes were interpolated using spherical spline interpolation, which is a widely used method in EEG analysis [1].' This would provide a brief justification of using the method and ensure the replicability.
Response 3: Thank you for the suggestions. It is important to report preprocessing methods rigorously. We have added relevant details about EEG Preprocessing:
EEG data were preprocessed using the functions in Curry8 software (Compumedics Neuroscan, Charlotte, NC, USA). First, the default reference electrode was set to the bilateral mastoid electrodes (M1 and M2). The missing electrodes were interpolated using spherical spline interpolation, where the sphere center of new electrodes is subtracted before the interpolation. It is a widely used method in EEG analysis [49]. Continuous EEG signals were band-pass filtered 0.1–30 Hz. Next, the detection range to 200 ms before and 200 ms after eye blinks was set to ocular artifact reduction. Epochs with amplitude values exceeding ± 100 μV at any electrode were excluded from the average. 16.17% of the trials were excluded due to artifacts (popular peer condition = 16.25%; rejected peer condition = 16.80%; neglected peer condition = 15.27%; unfamiliar peer condition = 16.34%). The data were segmented starting 200 ms before the onset of painful or nonpainful pictures and lasting until 800 ms after these onsets. EEG epochs were base-line-corrected by a 200 ms time interval before the target stimuli onset. Finally, EEGLAB v2021.1 [50] was used to calculate the trial average for each participant and condition.
[49] Kim, H.; Luo, J.; Chu, S.; Cannard, C.; Hoffmann, S.; Miyakoshi, M. ICA’s Bug: How Ghost ICs Emerge from Effective Rank Deficiency Caused by EEG Electrode Interpolation and Incorrect Re-Referencing. Front. Signal Process. 2023, 3. https://doi.org/10.3389/frsip.2023.1064138.
Comments 6: Regarding the subjective measures, I think relying solely on raw scores ignores individual response tendencies and can weaken the robustness of the findings. The authors should address the response bias issue which can distort results. Otherwise, comparisons across participants might reflect differences in rating styles rather than genuine differences in perceived empathy or recognition.
Response 6: We gratefully appreciate your valuable comment. Through an extensive review of the literature, we found that pain empathy could be evaluated using empathy questionnaires (Fusaro, Tieri, & Aglioti, 2016; Nazarewicz, Verdejo-Garcia, & Giummarra, 2015; Zhao et al., 2020). It could also be evaluated by subjective ratings and brain responses to visual stimuli depicting or indicating that another person is in pain (Cui, Zhu, & Luo, 2017; Decety, Yang, & Cheng, 2010; Fan & Han, 2008). Subjective measures are frequently used in research on pain empathy to assess participants' perceived pain intensity and the unpleasantness of the stimuli. The statistical analysis results of the raw scores from subjective measures are often presented in the Behavioral Results section (Ren et al., 2019; Wu et al., 2020; Meng et al., 2013; Yan et al., 2017). Besides, the painful and non-painful pictures used in our study were pre-evaluated, and there were significant differences between the painful and non-painful pictures. Moreover, participant rating style differences may exist, but our within-subject experimental design may help mitigate potential interindividual differences. We focused on the differences in pain ratings across different peer-type conditions for all participants.
ð These are good points. Please incorporate them into the discussion.
Response 6: Thank you for the suggestions. We have added supplementary content to the Discussion: Pain empathy could be evaluated using empathy questionnaires [55−57]. It could also be evaluated by subjective ratings and brain responses to visual stimuli depicting or indicating that another person is in pain [17, 58, 59]. In our study, both behavioral and electrophysiological results revealed significant differences between painful and non-painful pictures and peer status exerting a greater influence on children's cognitive empathy processes than on their affective empathy processes. Early electroencephalographic components N1 and N2 are posited to reflect the early automated affective sharing in empathy for pain, that is, affective empathy. Late components P3 and LPP are thought to reflect the cognitive evaluation of painful stimuli, which pertains to cognitive empathy. According to the results of our study, in-group bias and peer status primarily affect the top-down sustained attention processing and cognitive evaluation of pain empathy, consistent with the results of Mathur et al. ’s [60] study. Mathur et al. [60] used fMRI to investigate racial group biases and discovered that the pain of in-group and out-group members led to differences in the medial prefrontal cortex, highlighting the role of cognitive empathy in in-group favoritism. The absence of differences in stimulus type and peer type in early electroencephalographic components may be attributed to the high heterogeneity of N1 and N2 components in pain empathy research. A meta-analysis indicated that the results of N1 and N2 are susceptible to various influences during the experimental processes (e.g., experimental paradigms, materials, etc.). In contrast, later electroencephalographic components P3 and LPP are considered robust indicators of pain empathy [61]. (Page 12, Paragraph 3, Lines 486−491)
We tried our best to improve the manuscript and made some changes marked in red in the revised paper. We appreciate the editor’s and reviewers’ warm work and hope the correction will be approved.

Round 3
Reviewer 1 Report
Comments and Suggestions for Authors
As the authors used a commercial software, the process can be replicated, so the description should be fine for now.
However, I would recommend taking some time to study the fundamental principles of digital filtering, including filter types (e.g., Butterworth, Chebyshev, FIR), filter order, and design windows (e.g., Hamming, Kaiser).